# Hippocalcin-like 4, a neural calcium sensor, has a limited contribution to pain and itch processing

Christopher G. Alvaro[1][†], João M. Braz[1], Mollie Bernstein[1], Katherine A. Hamel[1], Veronica Craik[1], Hiroki Yamanaka[1,2], Allan I. Basbaum[1]*

1 Department of Anatomy, University of California, San Francisco, CA, United States of America,
2 Department of Anatomy and Neuroscience, Hyogo College of Medicine, Hyogo, Japan

† Deceased.
* allan.basbaum@ucsf.edu

**Data Availability Statement:** All relevant data are within the paper and its Supporting Information files.

## Abstract

Calcium binding proteins are expressed throughout the central and peripheral nervous system and disruption of their activity has major consequences in a wide array of cellular processes, including transmission of nociceptive signals that are processed at the level of the spinal cord. We previously reported that the calcium binding protein, hippocalcin-like 4 (Hpcal4), is heavily expressed in interneurons of the superficial dorsal horn, and that its expression is significantly downregulated in a TR4 mutant mouse model that exhibits major pain and itch deficits due to loss of a subpopulation of excitatory interneurons. That finding suggested that Hpcal4 may be a contributor to the behavioral phenotype of the TR4 mutant mouse. To address this question, here we investigated the behavioral consequences of global deletion of Hpcal4 in a battery of acute and persistent pain and itch tests. Unexpectedly, with the exception of a mild reduction in acute baseline thermal responses, Hpcal4-deficient mice exhibit no major deficits in pain or itch responses, under normal conditions or in the setting of tissue or nerve injury. Taken together, our results indicate that the neural calcium sensor Hpcal4 likely makes a limited contribution to pain and itch processing.

## Introduction

The dorsal horn of the spinal cord harbors complex and highly organized neuronal networks through which incoming peripheral sensory information is processed and transmitted to higher centers in the brain. Although major classes of dorsal horn excitatory and inhibitory neurons have been defined according to morphological [1], electrophysiological [2–5] and neurochemical [6] properties, whether they are engaged by different sensory modalities or respond to multiple stimuli remains the subject of debate. Indeed, both specialized and polymodal populations of neurons have been described in the dorsal horn. For example, ablation of somatostatin-expressing dorsal horn interneurons results in a complete loss of mechanical pain, whereas thermal responses are largely intact [7]. VGlut3-expressing interneurons also contribute selectively to mechanical pain processing [8]. By contrast, mice that no longer express the extracellular matrix protein Reelin have significantly greater heat-, but reduced

**Funding:** This study was supported by NIH R35 NS097306 (AIB; JB; MB; KH; VC); IRACDA K12GM081266 (CA) The funders had no role in study design, data collection and analysis, decision to publish, or preparation of the manuscript.

**Competing interests:** The authors have declared that no competing interests exist.

mechanical- and unchanged cold-pain sensitivity [9–11]. Other studies provided evidence of specificity in the processing of messages that produce itch. For example, gastrin releasing peptide neuron-ablated mice are insensitive to most pruritic agents, but exhibit normal responses to noxious stimuli [12].

Taken together the above studies illustrate that labeled lines, which selectively respond to different modalities of pain or itch, exist in the spinal cord. On the other hand, some electrophysiological studies described populations of spinal cord neurons that respond to multiple stimulus modalities [13–17], indicative of sensory input convergence. Furthermore, dorsal horn deletion of Neurokinin 1 receptor-expressing neurons, which are targeted by both primary afferent and interneuronal substance P, reduces both pain and itch transmission [18,19]. On a more global scale, our laboratory identified a population of excitatory interneurons in the dorsal horn that is critical for the full behavioral expression of both pain and itch [20]. Specifically, mice that no longer express the testicular orphan nuclear receptor 4 (TR4) are less responsive to mechanical and chemical algogens as well as to a wide array of pruritic agents. Interestingly, however, although they have normal withdrawal latencies to noxious heat, they have reduced responses in heat pain tests that involve supraspinal processing. Underlying these profound deficits was an aberrant pattern of primary afferent termination and, perhaps more importantly, a profound loss of excitatory interneurons in the superficial dorsal horn. Interestingly, the missing population of excitatory interneurons was concentrated in laminae I-II and was neurochemically heterogeneous, including Calbindin-, Grp-, VGLUT2- and Reelin-expressing interneurons, making it difficult to attribute a particular deficit to a particular gene or neuronal subpopulation. However, in the course of interrogating the Allen Mouse and Brain Atlas (www. brain-map.org) for other genes that are highly expressed in the superficial dorsal horn, and following this with expression studies in the TR4 mutant mouse, we identified the neuronal calcium sensory, hippocalcin-like 4 (Hpcal4) as among the most highly expressed genes in the superficial dorsal horn, Most importantly, Hpcal4 expression is greatly reduced in the TR4 mutant mice, which suggested that it predominates in excitatory interneurons.

Hippocalcin-like 4 (Hpcal4) belongs to the superfamily of visinin-like calcium binding proteins (CaBPs) and contributes to the inactivation of Cav2.1 channels [21]. By regulating the levels of intracellular calcium, CaBPs contribute to a wide variety of cellular processes, including the transmission of nociceptive messages. For example, mice deficient for calbindin-D (28K) have reduced nocifensive behaviors in the formalin test and in the acetic acid test of visceral pain [22]. Furthermore, spinal cord injection of inhibitors of calmodulin-dependent protein kinase II (CaMK II) significantly reduces capsaicin-evoked discharge of spinal nociceptive neurons [23]. Mice deficient in the DREAM/KChIP3/calsenilin transcription factor, another member of the neural calcium sensor superfamily, exhibit reduced mechanical, thermal and chemical pain responses in both neuropathic and inflammatory pain models [24]. The neuronal calcium sensor-1 (NCS-1) has also been implicated in paclitaxel-induced neuropathic pain [25,26]. Whether Hpcal4 contributes to nociceptive processing has never been studied. Here we report that despite widespread dorsal horn expression that predominates in excitatory interneurons, with the exception of a small reduction in acute heat pain responsiveness, Hpcal4 KO mutant and wild type mice exhibit similar behavioral responses in a variety of tests of acute and chronic pain and itch.

## Materials and methods

### Animals

The animal experiments were approved by the UCSF Institutional Animal Care and Use Committee and were conducted in accordance with the NIH Guide for the Care and Use of

Laboratory animals. TR4 mutant mice were generated as previously described [20]. Hippocalcin-like 4-deficient mice were purchased from the UC Davis KOMP Repository (Stain ID: Hpcal4[tm1(KOMP)Vlcg]) and C57BL/6 mice were purchased from The Jackson Laboratory.

## Behavioral analyses

For all behavioral tests, adult mice (8–10 weeks old) were first habituated for 1 hour in Plexiglas cylinders. For the Hargreaves reflex test of heat pain sensitivity [27], mice were placed in clear plastic chambers on a glass surface through which a radiant heat source was focused on the hindpaw. We measured latency to withdraw the paw. In another test of heat pain, the tail was dipped into a 52°C water bath and the withdrawal latency recorded. For the hot plate test, we recorded the latency to lick or flinch the hindpaws or to jump. To test mechanical responsiveness we placed mice into clear plastic chambers on a wire mesh grid and stimulated the hindpaw with graded von Frey filaments. Withdrawal thresholds were determined using the up-down method [28]. To test chemical pain, we made intraplantar injections of either 3.0 µg capsaicin (Sigma) or 5% formalin (Acros Organics) in 10 µl and recorded the time spent licking/biting the injected hindpaw over the next 5 and 60 min, respectively. The experimenter performing the behavioral testing was always blind to treatment (WT or Hpcal4 KO). All statistical analyses were performed with Prism (Graph Pad) and data are reported as mean +/- SEM. Student $t$-tests were used for single comparisons between two groups. Other data were analyzed using two-way ANOVA.

## Pruritogen-evoked scratching

At least 24 hours prior to testing, mice were shaved at the nape of the neck under isoflurane anesthesia. The following pruritogens were dissolved in saline and injected (100 µl, s.c.) into the neck: chloroquine (Sigma, 100 µg) and histamine (Sigma 500 µg). The mice were video recorded, and we counted scratching bouts that occurred during the first 30 minutes after injection.

## Complete Freund's Adjuvant (CFA)

We used the CFA model of chronic inflammation as described previously [27]. Briefly, CFA (Sigma) was diluted 1:1 with saline and vortexed for 30 minutes. When fully suspended, we injected 20 µl of CFA into one hindpaw. Mechanical withdrawal thresholds were measured with the von Frey test, before (baseline) and 3 and 5 days after the injection.

## Spared-nerve injury (SNI) model of neuropathic pain

The SNI model was produced as described previously [29]. Briefly, under isofluroane anesthesia, we ligated and transected two of the three branches of the sciatic nerve, leaving the tibial nerve intact. Von Frey mechanical thresholds were tested before (baseline) and 2 and 7 days after injury.

## *In situ* hybridization

Mice were anesthetized with 2.5% Avertin and perfused transcardially with 0.1 M phosphate buffered saline (PBS) followed by 4% formaldehyde in PB. Lumbar spinal cord tissue was dissected out, post-fixed in the same fixative for 4h, cryopreserved overnight in 30% sucrose in PBS and sectioned at 6µm on a cryostat. Sections were then processed for *in situ* as described previously [30]. After development of the sections were stained with hematoxylin-eosin.

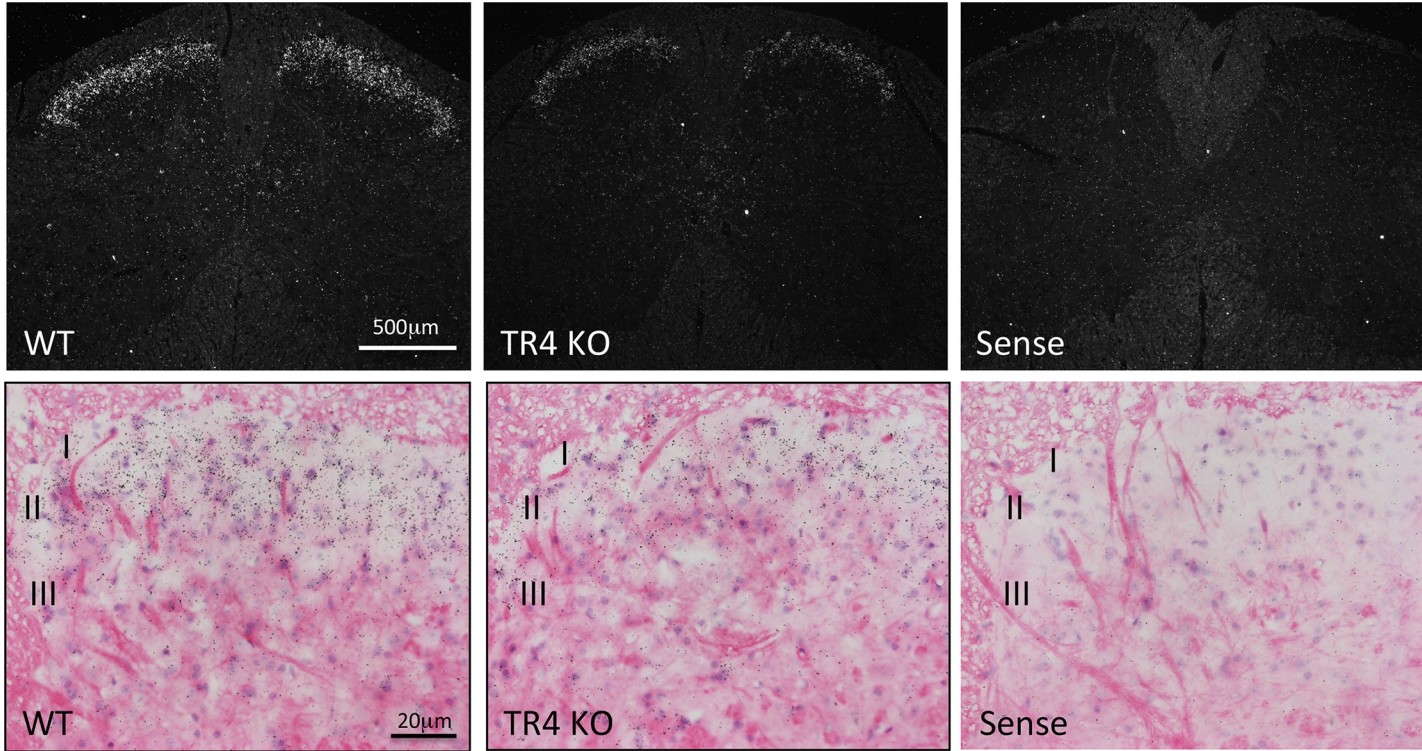

**Fig 1. Hippocalcin-like 4 (Hpcal4) expression in the spinal cord of naïve and TR4 mutant mice.** Darkfield (Top) and brightfield (bottom) panels illustrate situ hybridization expression patterns of Hpcal4 mRNA in the lumbar spinal cord of naïve (WT) and TR4 mutant (KO) mice. Hpcal4 is predominantly expressed in the substantia gelatinosa (laminae I-II) in naïve mice but strongly reduced in the spinal cord of TR4 mutant mice.

## Results

### Spinal cord expression of Hpcal4 in naïve and TR4 mutant mice

As noted above, a search in the Allen mouse Brain Atlas revealed that the Hpcal4 gene expression is concentrated in the most superficial laminae (I and II) of the mouse spinal cord. Using a radioisotope *in situ* hybridization protocol we confirmed that pattern of Hpcal4 expression in the adult mouse spinal cord (Fig 1). Hpcal4 expression predominated in the substantia gelatinosa (laminae I-II), where most nociceptors terminate. Importantly, Hpcal4 expression was strongly reduced in the spinal cord of TR4 mutant mice, suggesting that Hpcal4 is expressed by excitatory interneurons and that loss of Hpcal4 activity may have contributed to the behavioral deficits observed in the TR4 mutant mouse.

### Acute pain behaviors in Hpcal4-deficient mice

To address this question, we first examined the behavioral consequences of Hpcal4 deletion using a battery of acute pain tests. We found that baseline mechanical thresholds of Hpcal4 KO mice did not differ from naïve mice (Fig 2A). In contrast, we recorded mild, but significant thermal deficits in the Hpcal4 KO mice. This was true for both male and female Hpcal4 KO mice, which exhibited slightly higher paw withdrawal latencies in the Hargreaves test (Fig 2B). In the hotplate test, we found that male, but not female Hpcal4 KO mice exhibited higher latencies, at 48°C (Fig 2D). However, the differences disappeared at higher temperatures. On the other hand, we found no behavioral differences in the tail immersion tests for either gender (Fig 2C). Finally, nocifensive behaviors (licking/flinching) in response to hindpaw injection of

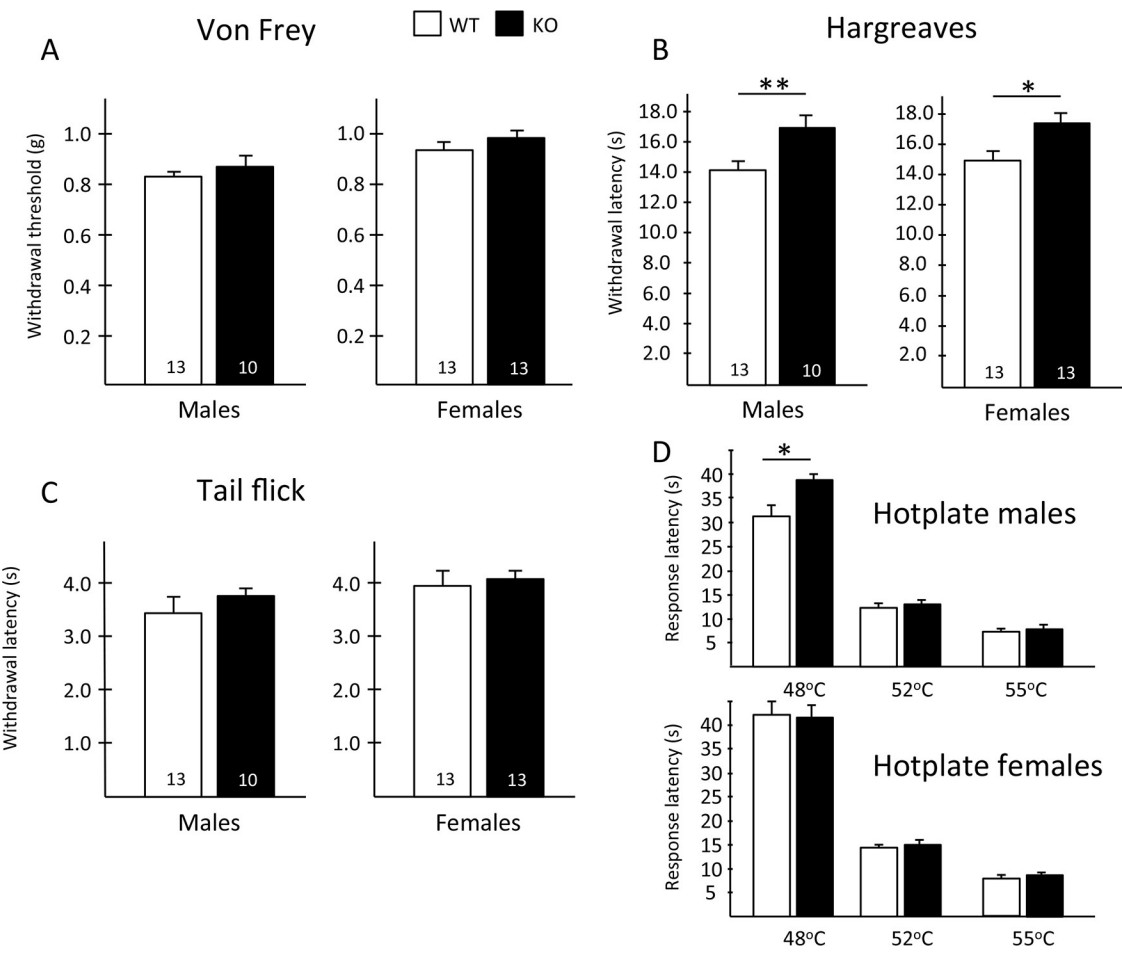

**Fig 2. Hippocalcin-like 4 mutant mice have reduced heat, but normal mechanical pain responses.** (A) Baseline mechanical thresholds of Hpcal4 mutant (KO, black bars) mice did not differ from naïve (WT, white bars) mice. (B) In contrast, both male and female Hpcal4 mutant mice exhibited higher paw withdrawal latencies in the Hargreaves test. (C) There is no behavioral differences in the tail immersion tests for either gender. (D) In the hotplate test male, but not female Hpcal4 mutant mice, exhibit higher response latencies, at 48˚C, but not at higher temperatures. Data are presented as mean ± SEM; Statistical significance was determined by Student's t-test in A-C and 2-way ANOVA (D), *p<0.05, **p<0.01.

capsaicin or formalin were similar in WT and Hpcal4 KO mice (Fig 3). Taken together, our results suggest that Hpcal4 modulates thermal, but not mechanical or chemical responsiveness in naïve conditions.

## Tissue and nerve injury-induced persistent pain

Next, we asked whether Hpcal4 contributes to the persistent pain that develops in the setting of tissue or nerve injury. Fig 4A and 4B illustrate that mechanical thresholds before and after hindpaw injection of CFA, to produce an inflammation model, did not differ between male or female WT and Hpcal4-deficient mice. In separate groups of mice, we performed the spared nerve injury model (SNI; [29]) and recorded mechanical thresholds before (baseline), 2 and 7 days after SNI. Fig 4C and 4D illustrate that male and female WT and Hpcal4-deficient mice exhibited comparable mechanical responsiveness before and after nerve injury. Taken together, these behavioral analyses indicate that Hpcal4 activity is not required for the development or maintenance of inflammation- or peripheral nerve injury-induced mechanical hypersensitivity.

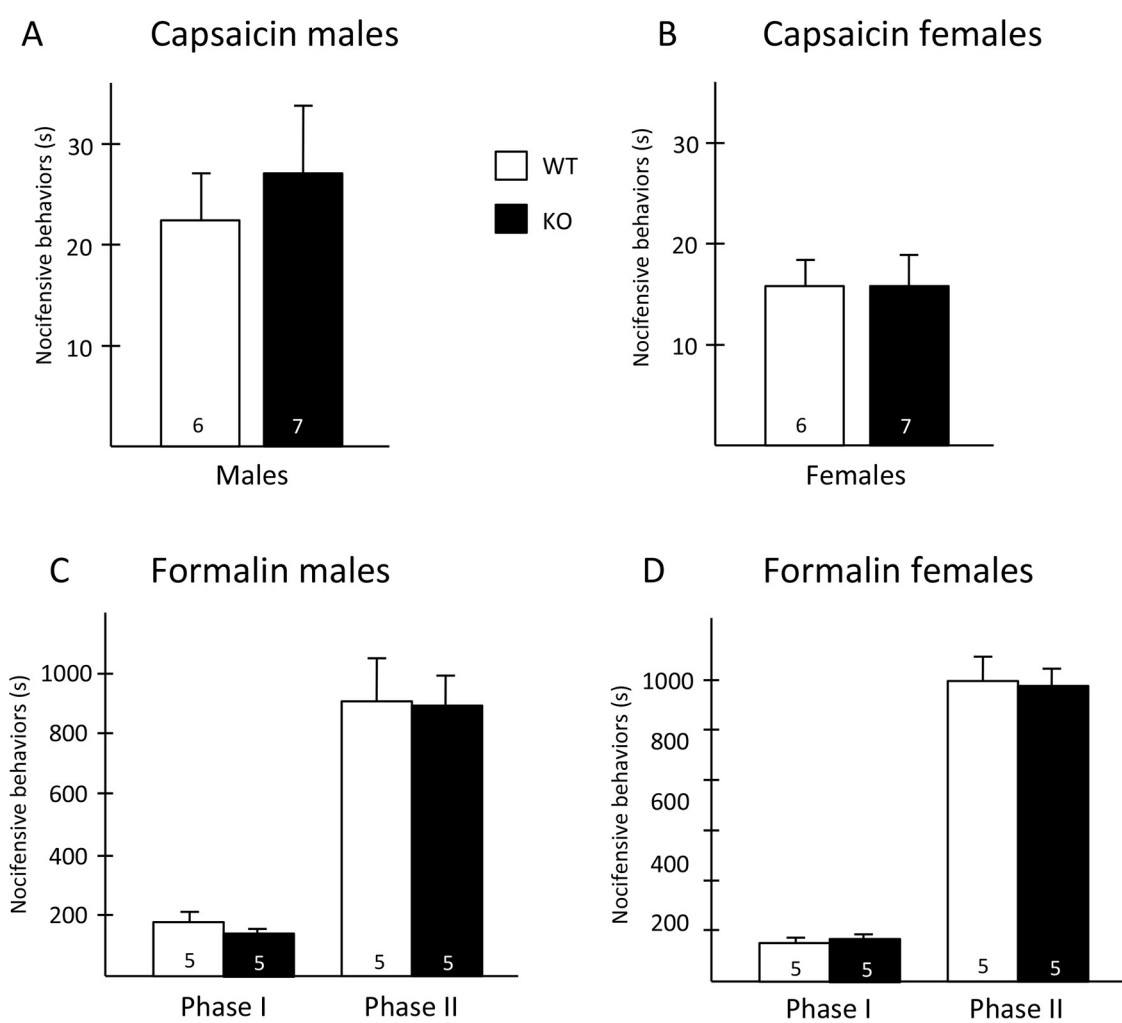

**Fig 3. Chemical pain responsiveness of Hippocalcin-like 4 mutant mice.** Nocifensive behaviors (licking/flinching) in response to hindpaw injection of capsaicin (A-B) or formalin (C-D) were similar in WT and Hpcal4 mutant mice. Data are presented as mean ± SEM. Statistical significance was determined by Student's t-test.

### Pruritogen-induced itch

Finally, we asked whether Hpcal4 expression is required to evoke scratching behavior in response to peripheral injection of pruritogens. Here we injected the nape of the neck of WT and Hpcal4 mutant mice with histamine (500 μg) or chloroquine (100 μg) and video recorded scratching behavior for the following 30 minutes. Fig 5 shows that both male and female Hpcal4 mutant mice exhibited the same levels of scratching bouts as did WT mice in response to both histamine (Fig 5A) and chloroquine (Fig 5B).

### Discussion

Hpcal4 belongs to the superfamily of calcium binding proteins [31], dysregulation of the activity of which negatively affects a wide variety of cellular processes, from homeostasis to learning and memory, to cancer and pain. For example, CaMKII-mediated phosphorylation of intracellular proteins contributes to spinal cord sensitization evoked by peripheral injections of capsaicin and inhibition of CaMKII reduces the responses of spinal cord nociceptive neurons [23].

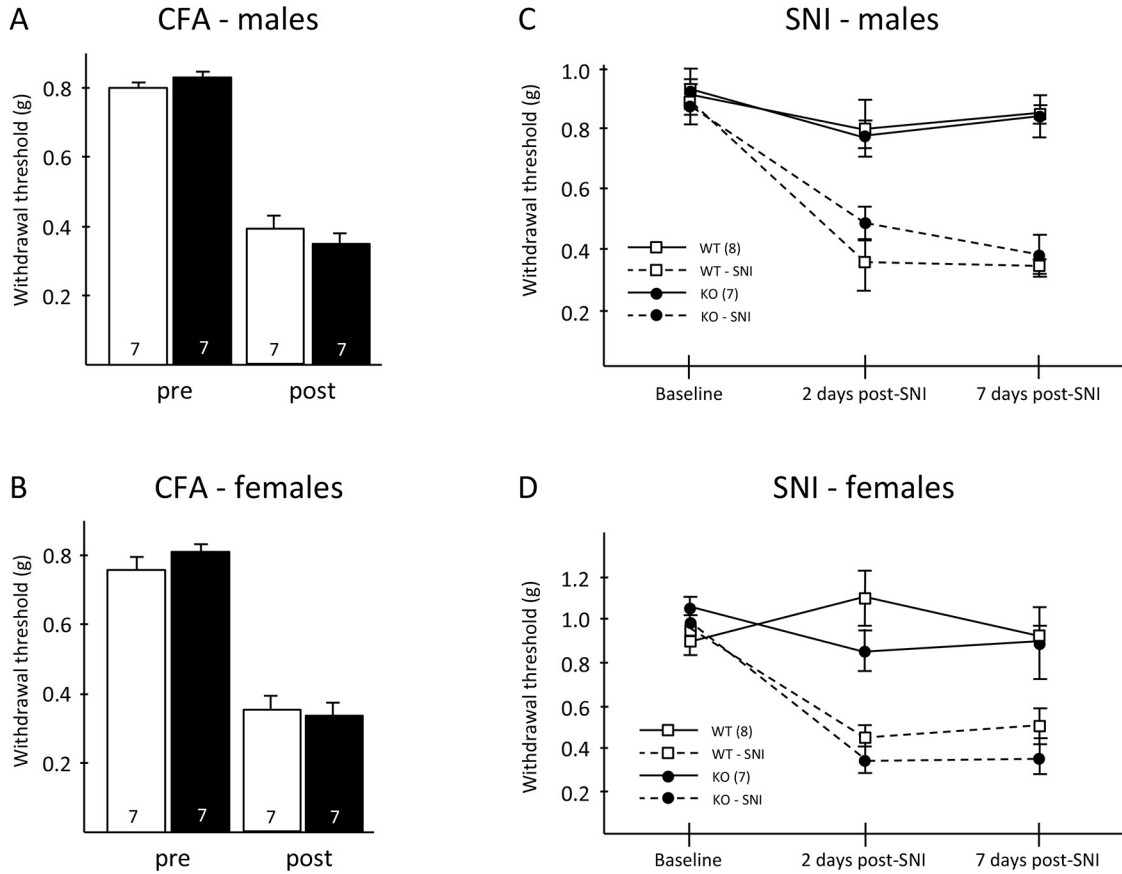

**Fig 4. Tissue and nerve injury-induced mechanical hypersensitivity in Hippocalcin-like 4 mutant mice.** (A-B) Mechanical allodynia did not differ between Hpcal4 mutant and control mice 3 days after CFA injection. (C-D) Hpcal4 mutant mice also developed normal mechanical allodynia 2 and 7 days after spared nerve injury (SNI). Data are presented as mean ± SEM. Statistical significance was determined by Student's t-test in A and B and by Two-way repeated measures ANOVA in C and D.

Inability of CamKII to autophosphorylate did not affect acute pain behaviors, but significantly reduced persistent pain behavior in the formalin test [32]. The possible involvement of Hpcal4 to pain and itch processing also relates to its expression pattern. In the spinal cord Hpcal4 is

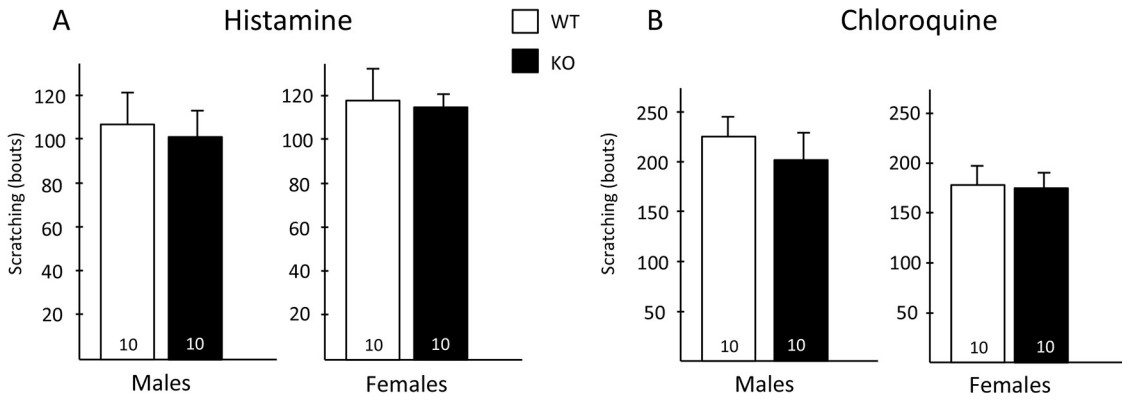

**Fig 5. Pruritogen responsiveness in Hippocalcin-like 4 mice.** Male and female Hpcal4 mutant mice respond normally (scratching bouts) to nape of the neck injection of histamine (A) or chloroquine (B). Data are presented as mean ± SEM. Statistical significance was determined by Student's t-test.

concentrated in neurons in laminae I-II of the superficial dorsal horn where most unmyelinated and lightly myelinated nociceptors terminate. Lastly, we demonstrated that CNS deletion of the TR4 gene in the mouse resulted in profound deficits in pain and itch responses due, in part to a developmental loss of excitatory interneurons in the superficial dorsal horn of the spinal cord [20]. Here we demonstrate that Hpcal4 is indeed among the genes that are significantly downregulated in the spinal cord of the TR4 mutant mice, which indicates that Hpcal4 is expressed, at least in part, in a subpopulation of excitatory interneurons. Based on a recent transcriptome analysis of dorsal horn neurons [33], it is likely that many of the Hpcal4 neurons that persist in the TR4 mutant mice are inhibitory.

Taking all these parameters into consideration, we hypothesized that Hpcal4 could contribute to pain and/or itch processing for several reasons. Surprisingly, however, our current behavioral analyses found little evidence in support of that hypothesis. We found that global and complete deletion of the Hpcal4 gene did not recapitulate the severe deficits of the TR4 mutant mice. In fact, except for acute heat responsiveness, the behavioral responses of Hpcal4 deficient mice were not altered in a wide variety of acute and persistent pain and itch tests. Note, however, that because we did not test the consequences of Hpcal4 deletion on thermal thresholds after CFA injection, we cannot rule out an effect on thermal hypersensitivity in the setting of inflammatory pain. Interestingly, similar to TR4 mutant mice, deletion of Hpcal4 resulted in significant decreased responses in the hotplate at 48˚C, but unlike TR4 deletion, not at higher temperatures. Curiously however, both male and female Hpcal4 mutant mice exhibited slightly but significantly higher thermal thresholds in the Hargreaves' test whereas TR4 male mutant mice do not. We did not determine whether any of the behavioral deficits in TR4 mutant mice were sex-dependent. Of course, only a conditional deletion approach can determine whether these discrete phenotypes arose from deletion of Hpcal4 from spinal cord neurons or from other CNS and/or peripheral neurons. Indeed, a recent single cell transcriptome analysis of primary sensory neurons [34] reported that some subtypes of unmyelinated sensory neurons expresses Hpcal4, notably the peptides and tyrosine hydroxylase-expressing subpopulations.

Taken together, we conclude that Hpcal4 has a limited contribution to spinal cord pain and itch processing both in naïve and tissue and nerve injury conditions. Whether compensations from Hpcal1 or other calcium binding proteins account for the lack of effect remains to be determined. Although very little is known about its contribution to pain and/or itch processing, Hpcal1 is indeed expressed in both excitatory and inhibitory interneurons of the dorsal and ventral horns of the spinal cord (cf. Allen Brain Atlas and [33]). Furthermore, the expression of Hpcal1 in TRPV1-expressing sensory neurons [33–35] suggests that this calcium neural sensor also contributes to calcium modulation in nociceptors.

## Supporting information

**S1 Data.**
(XLSX)

## Acknowledgments

We dedicate this manuscript to Cris Alvaro who sadly died soon after these studies were completed.

## Author Contributions

**Conceptualization:** Christopher G. Alvaro, João M. Braz, Allan I. Basbaum.

**Data curation:** Allan I. Basbaum.

**Formal analysis:** Christopher G. Alvaro, João M. Braz, Mollie Bernstein, Katherine A. Hamel, Veronica Craik, Hiroki Yamanaka, Allan I. Basbaum.

**Funding acquisition:** Allan I. Basbaum.

**Methodology:** Christopher G. Alvaro, João M. Braz, Mollie Bernstein, Katherine A. Hamel, Veronica Craik, Hiroki Yamanaka, Allan I. Basbaum.

**Project administration:** Christopher G. Alvaro, João M. Braz, Allan I. Basbaum.

**Resources:** João M. Braz, Allan I. Basbaum.

**Supervision:** Christopher G. Alvaro, João M. Braz, Allan I. Basbaum.

**Validation:** João M. Braz, Mollie Bernstein, Katherine A. Hamel, Veronica Craik, Hiroki Yamanaka, Allan I. Basbaum.

**Visualization:** Christopher G. Alvaro, João M. Braz, Mollie Bernstein, Katherine A. Hamel, Veronica Craik, Hiroki Yamanaka, Allan I. Basbaum.

**Writing – original draft:** João M. Braz, Allan I. Basbaum.

**Writing – review & editing:** João M. Braz, Allan I. Basbaum.

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
