## [Decision Letter · Decision Letter 0]

6 Nov 2019

PONE-D-19-29157

Contribution of the neural calcium sensor hippocalcin-like 4 to pain and itch processing

PLOS ONE

Dear Dr. Basbaum,

Thank you for submitting your manuscript to PLOS ONE. After careful consideration, we feel that it has merit but does not fully meet PLOS ONE’s publication criteria as it currently stands. Therefore, we invite you to submit a revised version of the manuscript that addresses the points raised during the review process.

We would appreciate receiving your revised manuscript by Dec 21 2019 11:59PM. To enhance the reproducibility of your results, we recommend that if applicable you deposit your laboratory protocols in protocols.io, where a protocol can be assigned its own identifier (DOI) such that it can be cited independently in the future. For instructions see: http://journals.plos.org/plosone/s/submission-guidelines#loc-laboratory-protocols

We look forward to receiving your revised manuscript.

Kind regards,

Karl-Wilhelm Koch, Ph.D.

Academic Editor

PLOS ONE

Journal Requirements:

1. In your data availability statement you write, "All relevant data are within the paper and its Supporting Information files." Please ensure you have provided the individual data points used to create the figures and determine means, medians and variance measures presented in the results, tables and figures (http://journals.plos.org/plosone/s/data-availability#loc-faqs-for-data-policy). If these data cannot be publicly deposited or included in the supporting information, e.g. due to patient privacy or ownership by a third party, explain why and explain how researchers may access them.

2. Thank you for including the following funding information within the acknowledgements section of your manuscript; "This work was supported by NIH R35 NS097306 and NS14627. "

Reviewers' comments:

Reviewer's Responses to Questions

**Comments to the Author**

1. Is the manuscript technically sound, and do the data support the conclusions?

Reviewer #1: Yes

Reviewer #2: Yes

2. Has the statistical analysis been performed appropriately and rigorously? 

Reviewer #1: Yes

Reviewer #2: Yes

3. Have the authors made all data underlying the findings in their manuscript fully available?

Reviewer #1: Yes

Reviewer #2: Yes

4. Is the manuscript presented in an intelligible fashion and written in standard English?

Reviewer #1: Yes

Reviewer #2: Yes

5. Review Comments to the Author

Reviewer #1: Neuronal Calcium Sensors (NCS) is a protein superfamily, itself, within the vast group of calcium binding proteins. At least two members of the NCS superfamily have been associated to pain perception: NCS-1 and DREAM/KChIP3/calsenilin. References to these studies are missing in the manuscript while the pain-related phenotype of calbindin-D deficient mice is cited, though this protein is only distantly related to NCS proteins.

Please correct/complete the name of all the authors in the same style: Yamanaka H

Reviewer #2: In the manuscript presented by Christopher GA et al, they investigated the behavioral effects of global deletion of the neural calcium sensor Hpcal4 in acute and persistent pain models. However, only a mild reduction in acute baseline thermal responses were observed in the mutant mice. And they show no major deficits in pain or itch responses in the setting of tissue or nerve injury. The conclusion that Hpcal4 makes a limited contribution to pain and itch processing was drawn.

The following questions need to be considered.

1. Usually, the SNI mouse model involves ligation of two of the three branches of the sciatic nerve, the tibial nerve and the common peroneal nerve, while the sural nerve is left intact. The authors used a different method, leaving the tibial nerve intact. What is the advantage of this approach?

2. Hpcal4-deficient mice showed lower heat sensitivity in the normal state and no alteration of mechanical hypersensitivity after CFA injection. How about the change of CFA-induced heat hypersensitivity? It is critical for the drawing the conclusion that Hpcal4-deficient mice exhibit no major deficits in inflammatory pain responses.

3. What is the functional relationship between TR4 and Hpcal4? The possible mechanisms leading to the reduced expression of Hpcal4 after TR4 gene deletion need to be discussed. In addition, the compensatory effect from Hpcal1 or other calcium binding proteins should be discussed as well.

4. Whether Hpcal4 is expressed in DRG neurons? Its expression pattern in DRG should be explained and discussed.

5. As demonstrated by the authors, Hpcal4 makes a limited contribution to pain and itch processing. Thus, the title of the manuscript should be more clear and more restricted.

6. PLOS authors have the option to publish the peer review history of their article (what does this mean?). If published, this will include your full peer review and any attached files.

Reviewer #1: Yes: Jose R Naranjo

Reviewer #2: Yes: Ying Zhang, Neuroscience Research Institute, Peking University, Beijing, China

---

## [Author Response · Author response to Decision Letter 0]

21 Nov 2019

Editorial requirements:

1. In your data availability statement you write, "All relevant data are within the paper and its Supporting Information files." Please ensure you have provided the individual data points used to create the figures and determine means, medians and variance measures presented in the results, tables and figures 

Response: We have uploaded an Excel file with the raw data.

2. Thank you for including the following funding information within the acknowledgements section of your manuscript; "This work was supported by NIH R35 NS097306 and NS14627. " We note that you have provided funding information that is not currently declared in your Funding Statement. However, funding information should not appear in the Acknowledgments section or other areas of your manuscript. We will only publish funding information present in the Funding Statement section of the online submission form. Please remove any funding-related text from the manuscript and let us know how you would like to update your Funding Statement. Currently, your Funding Statement reads as follows: "The funders had no role in study design, data collection and analysis, decision to publish, or preparation of the manuscript."

Response: We have removed funding-related text from the revised manuscript.

Reviewer #1: 

Neuronal Calcium Sensors (NCS) is a protein superfamily, itself, within the vast group of calcium binding proteins. At least two members of the NCS superfamily have been associated to pain perception: NCS-1 and DREAM/KChIP3/calsenilin. References to these studies are missing in the manuscript while the pain-related phenotype of calbindin-D deficient mice is cited, though this protein is only distantly related to NCS proteins. 

Response: We thank the Reviewer for pointing this out. Indeed, we were remiss not to include these studies in our manuscript. Reference to these studies has now been included in the Introduction section.

Please correct/complete the name of all the authors in the same style: Yamanaka H

Response: The name has been corrected.

Reviewer #2: 

In the manuscript presented by Christopher GA et al, they investigated the behavioral effects of global deletion of the neural calcium sensor Hpcal4 in acute and persistent pain models. However, only a mild reduction in acute baseline thermal responses was observed in the mutant mice. And they show no major deficits in pain or itch responses in the setting of tissue or nerve injury. The conclusion that Hpcal4 makes a limited contribution to pain and itch processing was drawn. The following questions need to be considered.

1. Usually, the SNI mouse model involves ligation of two of the three branches of the sciatic nerve, the tibial nerve and the common peroneal nerve, while the sural nerve is left intact. The authors used a different method, leaving the tibial nerve intact. What is the advantage of this approach?

Response: There are, in fact, different methods to generate the spared nerve injury model. In our approach, we ligate and cut the sural and peroneal nerves but leave the tibial intact. This results in profound and long lasting mechanical allodynia, which we described in detail in our Shields et al (2003) manuscript that reported an adaptation of the DeCosterd et al rat model. There are some advantages to our approach. Namely, by leaving the tibial nerve intact, we have access to a much bigger area of the footpad to apply the von Frey filaments and there is considerably less motor impairment.

2. Hpcal4-deficient mice showed lower heat sensitivity in the normal state and no alteration of mechanical hypersensitivity after CFA injection. How about the change of CFA-induced heat hypersensitivity? It is critical for the drawing the conclusion that Hpcal4-deficient mice exhibit no major deficits in inflammatory pain responses.

Response: As the reviewer notes, we did not analyze the consequences of Hpcal4 deletion on thermal thresholds after CFA and therefore we cannot rule out an effect on thermal hypersensitivity in the setting of inflammatory pain. As thermal hypersensitivity cannot reliably be generated in the SNI, partial peripheral nerve injury models (Shields et al., 2003), we decided to focus our analyses on the mechanical changes that occur after nerve and tissue injuries. We have included these considerations in the Discussion section of the revised manuscript.

3. What is the functional relationship between TR4 and Hpcal4? The possible mechanisms leading to the reduced expression of Hpcal4 after TR4 gene deletion need to be discussed. In addition, the compensatory effect from Hpcal1 or other calcium binding proteins should be discussed as well.

Response: As we previously demonstrated, deletion of the orphan nuclear receptor TR4 results in profound defects in a wide variety of pain and itch responses that are associated with a developmental loss of excitatory interneurons in the superficial dorsal horn of the spinal cord. We presume that the Hpcal4 reduction observed in the TR4 deficient mouse results from the loss of this population of excitatory neuron, rather than the loss of a functional relationship between TR4 and Hpcal4. In fact, we have no evidence that TR4 and Hpcal4 are at all related. On the other hand, we have now discussed in the revised manuscript the possible compensatory effects provided by Hpcal1.

4. Whether Hpcal4 is expressed in DRG neurons? Its expression pattern in DRG should be explained and discussed.

Response: To our knowledge, there are no neuroanatomical (immunocytochemical or in situ) studies that analyzed the expression pattern of Hpcal4 in sensory neurons. However, a recent transcriptome analysis did suggest that both Hpcal1 and 4 are expressed in subpopulations of DRG neurons. These considerations have been now added in the Discussion of the revised manuscript. Of course, we recognize that because this was a complete null, that there is always a possible contribution of the Hpcal4 loss in areas other than the dorsal horn.

5. As demonstrated by the authors, Hpcal4 makes a limited contribution to pain and itch processing. Thus, the title of the manuscript should be more clear and more restricted.

Response: We have modified the title of the manuscript to better reflect our conclusions.

---

## [Editor Report · Decision Letter 1]

25 Nov 2019

Hippocalcin-like 4, a neural calcium sensor, has a limited contribution to pain and itch processing

PONE-D-19-29157R1

Dear Dr. Basbaum,

We are pleased to inform you that your manuscript has been judged scientifically suitable for publication and will be formally accepted for publication once it complies with all outstanding technical requirements.

With kind regards,

Karl-Wilhelm Koch, Ph.D.

Academic Editor

PLOS ONE
---

## [Editor Report · Acceptance letter]

23 Jan 2020

PONE-D-19-29157R1 

Hippocalcin-like 4, a neural calcium sensor, has a limited contribution to pain and itch processing 

Dear Dr. Basbaum:

I am pleased to inform you that your manuscript has been deemed suitable for publication in PLOS ONE. Congratulations! Your manuscript is now with our production department. 

With kind regards,

on behalf of

Dr. Karl-Wilhelm Koch 

Academic Editor

PLOS ONE